# Effects of aircraft noise exposure on self-reported health through aircraft noise annoyance: Causal mediation analysis in the DEBATS longitudinal study in France

**Minon'tsikpo kossi KODJI**[1]*, **Lise GIORGIS-ALLEMAND**[1], **Sylviane LAFONT**[1], **Émilie LANOY**[2,3,4], **Anne-Sophie EVRARD**[1]

**1** Univ Lyon, Univ Gustave Eiffel, Ifsttar, Univ Lyon 1, Umrestte, UMR T_9405, Bron, France, **2** AP-HP, Hôpital Européen Georges Pompidou, Unité de Recherche Clinique, APHP Centre, Paris, France, **3** Inserm, Centre d'Investigation Clinique 1418 (CIC1418) Epidémiologie Clinique, Paris, France, **4** HeKA, INRIA PARIS, Université de Paris, Paris, France

* minon.kodji@univ-eiffel.fr

## Abstract

### Background

Previous studies reported an association between transportation noise and self-reported health status (SRHS). They also suggested a mediating role of noise annoyance using conventional statistical methods. These methods are subject to bias in longitudinal studies with time-dependent exposure, mediator and confounding factors. This study aims to investigate the mediating role of aircraft noise annoyance in the effect of aircraft noise on SRHS using a causal inference approach to address time-dependent variables issues.

### Methods

We used data from 881 participants in all three visits in the DEBATS longitudinal study conducted around three French airports. Participants over 18 years of age reported their self-perceived health status, aircraft noise annoyance, and noise sensitivity by completing a questionnaire at three visits in 2013, 2015 and 2017. Noise maps were used to estimate aircraft noise levels outside their homes. Marginal structural models with inverse probability weighting were used to estimate the total effect of aircraft noise levels on SRHS and its decomposition into direct and indirect effect through aircraft noise annoyance.

### Results

This study showed a deleterious effect of aircraft noise on SRHS. The odds ratio (OR) corresponding to the total effect and comparing the highest aircraft noise category ($\geq$60 dBA) to the reference category (<50 dBA) was significant ($OR_{poor/fair\_SHRS}$ = 1.25 (95%CI: 1.06 to 2.08)). It also showed no direct effect of aircraft noise levels on SRHS, but an indirect effect through annoyance. This indirect effect increased as aircraft noise levels increased, with a statistically significant OR when comparing the highest noise category ($\geq$60 dBA) to the

**Data Availability Statement:** The data used in this analysis contain sensitive and identifying personal information from participants in France. The

participants study did not provide consent that their data be made public and permission to do so has not been granted by the French personal protection committee (CPP) and the French National Commission for Information Technology and Liberties (CNIL). Gustave Eiffel University has a data sharing policy and the authors will make the data available upon request to qualified researchers working within institutions with evidence that they comply with current GDPR ethical and professional standards and requirements if all local participating centers are able to gain relevant permissions from the French personal protection committee (by completing the request form via https://siriph.sante.gouv.fr) and from the French National Commission for Information Technology and Liberties (by completing the request form via https://declarations.cnil.fr/declarations/declaration ; Address: 3 Place de Fontenoy TSA 80715 75334 PARIS CEDEX 07 France ; Telephone : +33 1 53 73 22 22). Please contact the corresponding author or the Data Protection Officer of the Gustave Eiffel University for access to the data. Contact information: by e-mail (protectiondesdonnees-dpo@univ-eiffel.fr) or by post (Data Protection Officer, Université Gustave Eiffel - Campus de Marne-la-Vallée, 5 Boulevard Descartes, 77420 Champs sur Marne, France).

**Funding:** The present study was supported by funds from the French Ministry of Health, the French Ministry of the Environment, the French Civil Aviation Authority, the Airport Pollution Control Authority (Acnusa) and the Gustave Eiffel University. The authors would like to thank them for their kind assistance.

**Competing interests:** The authors have declared that no competing interests exist.

lowest (<50 dBA) ($OR_{poor/fair\_SHRS}$ = 1.16 (95%CI: 1.03 to 1.52)). Nearly 66% of aircraft noise's effect on SRHS was mediated by aircraft noise annoyance.

## Conclusion

This study supports the deleterious causal effect of aircraft noise on SRHS. The results highlight the important mediating role of aircraft noise annoyance in the causal pathway from exposure to aircraft noise to poor/fair SRHS.

## Introduction

Environmental exposure to transportation noise affects the health of a large number of people [1]. Investigating the causal effects of noise on health from observational studies is challenging and requires longitudinal data on exposure, confounders and health outcomes. Previous studies found an association between aircraft noise levels and self-reported health status (SRHS) [2–6]. Part of the observed deleterious effect of aircraft noise on SRHS could be indirect through aircraft noise annoyance: increasing aircraft noise levels induce aircraft noise annoyance [7], which in turn negatively affects health [8, 9]. SRHS is widely used to measure the general health of populations. This is a multi-dimensional indicator that considers health and lifestyle practices, functional, coping and well-being dimensions [10, 11]. It has been shown to be a good predictor of quality of life, morbidity and mortality [12, 13]. In observational settings, confounders imply structural relationships between aircraft noise levels, aircraft noise annoyance and SRHS and thus could prevent the identification of the causal effect of aircraft noise levels on SRHS with aircraft noise annoyance as a mediator [14]. For example, confounding factors exist when a variable is associated with both increased noise annoyance and poor/fair SRHS.

Some previous studies have examined the relationship between transportation noise and health issues using structural equation models (SEMs) [8, 15–20]. Four longitudinal studies have evaluated the causal effect of noise annoyance on psychological factors such as noise sensitivity, fear of noise source, personal dependency on noise source, belief that noise can be avoided [16], on some health outcomes such as mental health well-being [8] and depression [20], and on trust in authorities [19]. The results showed a reverse causality of the outcomes mentioned above on noise annoyance. Pre-existing mental health conditions could result in higher noise annoyance.

Standard mediation analysis approaches, such as the Baron and Kenny approach [21] or SEMs [22], are often used in the context of one time point measurement of exposure, mediator, confounders and outcome, but do not address issues of time-varying exposure, mediator and confounding factors. These methods may lead to biased estimates when the confounders and outcome at a given time point are affected by prior exposure in longitudinal settings [23]. Assessing the causal effect of aircraft noise levels on health from longitudinal observational data requires adequate control of all time-dependent variables to limit confounding bias. In the presence of a mediator, the total effect of aircraft noise on SRHS includes the direct effect of aircraft noise on SRHS that is not mediated by aircraft noise annoyance and the indirect effect of aircraft noise on SRHS that is mediated by aircraft noise annoyance. Identifying all these effects requires assumptions on unmeasured confounders [14] including in time-varying settings, the absence of confounders in the causal effect of the mediator -annoyance- on the outcome -SRHS- affected by the prior exposure -noise [24]. For circumventing this latter

assumption, VanderWeele and Tchetgen Tchetgen introduce a randomized interventional effect similar to direct and indirect effects based on inverse probability weighting (IPW) mimicking a randomized trial for better control of bias [25].

In this study, our objective was to determine, the total effect of aircraft noise on SRHS, the respective importance of the effect mediated by noise annoyance, and of the one not mediated by noise annoyance, using this causal mediation analysis approach.

## Methods

### Study population

The DEBATS (Discussion on the health effects of aircraft noise) longitudinal study included in 2013 (T0) 1,244 residents living near three major French airports (Paris-Charles de Gaulle, Lyon Saint-Exupéry, and Toulouse-Blagnac). The study followed up the residents in 2015 (T1) and in 2017 (T2). For each visit, participants were invited to complete a questionnaire during a face-a-face interview at their home with an interviewer. The questionnaire was designed to assess their self-reported health status, aircraft noise annoyance, sensitivity to noise, demographic and socio-economic characteristics and lifestyle. More details on the study design are available in the article by Kourieh et al [26].

The present study was limited to the 881 participants who attended all three visits (inclusion in 2013 (T0), first follow-up in 2015 (T1) and second follow-up in 2017 (T2)).

Two national authorities in France, the French Advisory Committee for Data Processing in Health Research (CCTIRS 11–405) and the French National Commission for Data Protection and the Liberties (DR 2012–361) approved the present study. The participants signed and returned an informed consent form by mail.

### Aircraft noise exposure

Aircraft noise levels were estimated outside participants' homes. For this purpose, a geographic information system was used to link participants' addresses to aircraft noise levels from noise maps produced by Paris airports and the French Civil Aviation Authority using the integrated noise model [27]. For Paris-Charles de Gaulle airport, aircraft noise levels could be estimated for each of the three visits using noise maps from 2013, 2015 and 2017. However, for Lyon-Saint-Exupéry and Toulouse-Blagnac airports, only the noise maps from 2003 and 2004 respectively were available. For these two airports, the same estimated aircraft noise levels were considered for all three visits. In this study, we used the $L_{den}$ energy indicator based on energy equivalent noise level over a whole day [6.00–18.00] with an additional penalty of 10 dBA for night time noise [in France: 22.00–6.00] and an additional penalty of 5 dBA for evening noise [in France: 18.00–22.00] [28]. Aircraft noise levels were considered in four categories in the statistical analyses: <50, 50 to 54, 55 to 59, >60 dBA.

### Self-rated health status

In the questionnaire, participants self-reported their health status (SRHS) by answering the following question: "In general, would you say that your health is excellent, good, fair, or poor?" Participants who reported fair or poor health were compared to those who reported good or excellent health.

### Aircraft noise annoyance

Participants assessed aircraft noise annoyance by answering the following ICBEN 5-point scale question [29]: "Thinking about the last 12 months, when you are here at home, how much

does aircraft noise bother, disturb, or annoy you? Possible responses were: "extremely", "very", "moderately", "slightly" or "not at all". These five categories of aircraft noise annoyance were considered in the main analysis. In a sensitivity analysis, following ISO/TS recommendations [30], two categories -"extremely" or "very" versus "moderately," "slightly," or "not at all"- of aircraft noise annoyance corresponding to highly annoyed versus not highly annoyed participants were considered.

## Confounders

Potential confounders of the effect of aircraft noise levels on aircraft noise annoyance, of aircraft noise annoyance on SRHS and of aircraft noise levels on impaired SRHS were included in the models [5, 6, 31]. Time-varying confounders measured at all visits were used, including age (in six categories: 18 to 34, 35 to 44, 45 to 54, 55 to 64, 65 to 74 and $\geq$ 75 years old), the number of people in the household (in four categories: 1, 2, 3 or $\geq$ 4 people), smoking habits (in three categories: ex-smoker, occasional/daily smoker, non-smoker), alcohol consumption (in three categories: never, light drinker, moderate or heavy drinker), sport activities (yes/no), the income per consumption unit (CU) (in three categories: <1550, 1550 to 2750, >2750 euros per month), noise sensitivity (in two categories: highly sensitive to noise, not highly sensitive to noise), roof or window insulation (yes/no), and satisfaction with the living environment characterized by a neighborhood attachment score. This score derived from four standardized questions about participants' attachment to their neighborhood (from 0 to 10: the higher the score, the more satisfied with their living environment) [31]. From the household income range reported by the participants in the questionnaire, the income per CU was then calculated using the average of each income category divided by CU (according to the Insee definition of CU [32], the first adult counts as 1 CU, other people aged 14 or over as 0.5 CU and children under 14 as 0.3 CU). Noise sensitivity was assessed from the following question: "Regarding noise in general, compared to people around you, do you think that you are much less sensitive than, or less sensitive than, or as sensitive as, or more sensitive or much more sensitive than people around you?". Participants who reported being 'more' or 'much more sensitive than people around them' were considered to be highly sensitive to noise. Participants who reported being 'much less sensitive' or 'less sensitive' or 'as sensitive as people around them' were considered to be not highly sensitive to noise. Window or roof insulation was self-reported by participants in response to the question "Does your dwelling have roof (or window) insulation against external noise?" The dwelling was considered to be insulated if at least one of the roof or windows was insulated. Static confounders such as sex, education (three categories: < baccalaureate, baccalaureate to baccalaureate + 2, $\geq$ baccalaureate + 3) and country of birth (France versus other country) were also collected at baseline and included in the models. The interviewer recorded the sex based on the following question, which was addressed to the interviewer and not the participant: "What is the sex of the person responding to the questionnaire? Female or Male?" Pre-existing diseases were included in sensitivity analysis. This variable corresponded to the presence of at least one comorbidity diagnosed in the last 12 months prior to the interview, from among the following: diabetes, cardiovascular diseases, myocardial infarction, cancer, diagnosed hypertension and medication use.

## Statistical analysis

Mediation analysis with time varying exposure, mediator, outcome and confounders was conducted using marginal structural models with inverse probability weighting [25]. Model assumptions on causal effects of time-varying exposure (aircraft noise levels), time-varying confounders, time-varying mediator (aircraft noise annoyance) and time-varying outcome

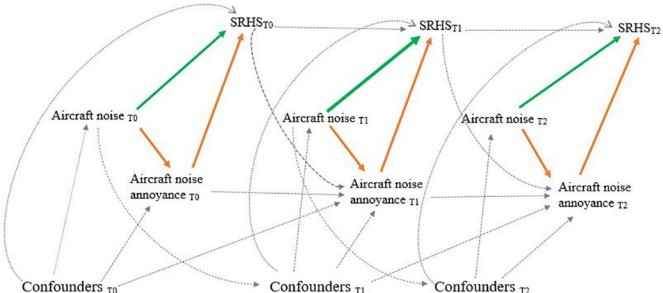

**Fig 1. Causal graph with time-varying aircraft noise levels, time-varying confounders, time-varying aircraft noise annoyance and time-varying self-reported health status.** Green arrows for direct effect, orange arrows for indirect effect and dotted arrows for confounding effect. Some arrows were omitted when no direct effect from a variable on another was hypothesized.

(SRHS) were illustrated using a longitudinal directed acyclic graph (DAG) (Fig 1). These hypothesized relationships were tested using SEMs. The total effect and its decomposition into direct effect and indirect effect were estimated. The mediated proportion was estimated by dividing the indirect effect by the total effect. The effect of aircraft noise levels during the three visits on $SRHS_{T2}$ measured at the end of the follow-up was evaluated. Intermediate SRHS values $SRHS_{T0}$ and $SRHS_{T1}$ were considered as confounders at T1 and T2 respectively and used for weights estimation to consider potential reverse causation.

Let $A(t)$, $M(t)$, $Y(t)$, $L(t)$, t = 0,1,..,$T$, denote the values of aircraft noise levels, aircraft noise annoyance, SRHS and confounders respectively measured at the time t, and $\overline{a}(a(0), a(1), a(2))$ the history of exposure at the three visits T0, T1 and T2. The three interventional effects were estimated by combining two marginal structural models' parameters [25]: 1) the first marginal structural model (M1 model) simultaneously assessed the cumulative effect of the exposure (aircraft noise levels) and the cumulative effect of the mediator (aircraft noise annoyance) on the outcome at the end of follow-up; 2) the second marginal structural model assessed for each time $t$, the effect of the exposure (aircraft noise levels) on the mediator (aircraft noise annoyance) at time t:

$$log\{\mathbb{E}[Y_{\overline{am}}]\} = \theta_0 + \theta_1 cum(\overline{a}) + \theta_2 \, cum(\overline{m}) \qquad (M1)$$

$$log\{\mathbb{E}[M_{\overline{a}}(t)]\} = \beta_0(t) + \beta_1(t) \, avg\,(\overline{a}\,(t)) \qquad (M2)$$

where $cum(\overline{a}) = \sum_{t \leq T} a(t)$ and $avg\,(\overline{a}\,(t))$ the average noise level from the inclusion to the time t. $\theta_1$ and $\theta_2$ represent the cumulative effect of aircraft noise levels and the cumulative effect of aircraft noise annoyance on SRHS, respectively. $Y_{\overline{am}}$ is a counterfactual variable that would be observed if an individual were exposed to a history of aircraft noise levels $\overline{a}$ and with an annoyance history $\overline{m}$. Under three identifiability assumptions (consistency, no unmeasured confounders for exposure and positivity) [33], marginal models' parameters were estimated consistently using inverse probability weighting (IPW), which corrects for cumulative confounding over time. The IPW allowed us to estimate the causal effect of aircraft noise levels on SRHS by emulating a hypothetical randomized experiment through the creation of a pseudo-population in which exposed and non-exposed subjects were exchangeable within levels of the available confounders. Formally, if the $i^{th}$ individual is observed to have $a_i(t)$, $m_i(t)$ as aircraft noise and aircraft noise annoyance (mediator) levels at time t, the stabilized time-

dependent weights of exposure and mediator were estimated as follows [34]:

$$w_i^A(t) = \frac{P\{A_i(t) = a_i(t) \mid \overline{a}_i(t-1), \overline{m}_i(t-1)\}}{P\{A_i(t) = a_i(t) \mid \overline{a}_i(t-1), \overline{m}_i(t-1), \overline{l}_i(t)\}}$$

$$w_i^M(t) = \frac{P\{M(t) = m_i(t) \mid \overline{a}_i(t), \overline{m}_i(t-1)\}}{P\{M(t) = m_i(t) \mid \overline{a}_i(t), \overline{m}_i(t-1), \overline{l}_i(t)\}}$$

The denominator of the weight $w_i^A(t)$ corresponds to the conditional probability that the ith individual will be exposed to his/her observed aircraft noise levels at time t, given past values of aircraft noise levels, annoyance and confounders. The numerator-stabilizer consists in multiplying by the marginal probability of exposure which does not depend on time-dependent confounding factors and enables more efficient estimation in decreasing the variance. Analogous definition was retained for mediator weight $w_i^M(t)$. The weight $\prod_{t=0}^{T} w_i^A(t) \times w_i^M(t)$ was used to estimate the parameters of M1 model and $\prod_{l=0}^{t} w_i^A(l)$ to estimate the parameters of M2 model. Of the regression models' specifications for weights estimation including a potential interaction, the one with the mean weight closest to 1 with the lowest standard error was selected. The final models included interaction between aircraft noise levels and noise sensitivity and between noise levels and sex.

For marginal M1 and M2 models, sensitivity analyses were conducted: i) using marginal M3 and M4 models in which the joint effect of aircraft noise levels history $\overline{a}$ and the joint effect of aircraft noise annoyance history $\overline{m}$ on poor/fair SRHS were not supposed to be cumulative (parameters depend on follow-up time t):

$$log\{\mathbb{E}[Y_{\overline{am}}]\} = \theta_0 + \sum_{t \leq T} \theta_1(t).a(t) + \sum_{t \leq T} \theta_2(t).m(t) \tag{M3}$$

$$log\{\mathbb{E}[M_{\overline{a}}(t)]\} = \beta_0(t) + \sum_{l \leq t} \beta_1(l).a(l) \tag{M4}$$

ii) using two categories for the mediator -aircraft noise annoyance; iii) using truncated weights at the 5th and 95th percentiles to explore potential biases that individuals with extreme weights might introduce into the estimates.

For all regression models, the effect size of a given exposure was provided as the odds ratio for poor/fair SHRS noted $OR_{poor/fair\_SHRS}$. The 95% confidence intervals were derived from non-parametric bootstrap with 7,500 resamples to ensure a sufficient number of estimates in case of non-convergence [35]. Bootstrap estimation correction approaches (Bca) were used to control estimation bias due to non-symmetrical bootstrap distributions [35]. All analyses were performed using SAS 9.4.

## Results

Table 1 presents the characteristics of the 811 participants (373 men and 438 women) who attended all three visits (T0, T1 and T2). At baseline, about 14% of participants reported poor or fair health. This proportion remained similar over the three visits (p-value = 0.69). The number of participants in each category of aircraft noise was relatively equal (around 25%) at baseline (T0), as a result of the stratification used to select the study population. The proportion of participants in the < 50 dBA category increased at both T1 and T2 follow-ups (28% at T1 and 30% at T2 respectively). At baseline, 18% of participants reported being highly annoyed by aircraft noise. This proportion then increased (25% at both T1 and T2; p-value < 0.01). About 30% of participants reported being highly sensitive to noise during all three visits.

**Table 1. Characteristics of the 811 participants who attended all three visits.**

| Variables | T0 | T1 | T2 | p-value[1] |
|---|---|---|---|---|
| SRHS | | | | 0.48 |
| Poor/ fair | 113 (13.9) | 123 (15.2) | 112 (13.8) | |
| Good/excellent | 698 (86.1) | 688 (84.8) | 699 (86.2) | |
| Noise levels ($L_{den}$ in dBA) | | | | 0.33 |
| <50 | 209 (25.8) | 230 (28.4) | 242 (29.8) | |
| 50–54 | 215 (26.5) | 211 (26.0) | 208 (25.7) | |
| 55–59 | 195 (24.0) | 185 (22.8) | 180 (22.2) | |
| $\geq$ 60 | 192 (23.7) | 185 (22.8) | 181 (22.3) | |
| Aircraft noise annoyance | | | | < 0.01 |
| Extremely | 25 (3.1) | 37 (4.6) | 40 (4.9) | |
| Very | 121 (14.9) | 165 (20.4) | 161 (19.9) | |
| Moderately | 204 (25.1) | 308 (38.0) | 281 (34.6) | |
| Slightly | 314 (38.7) | 190 (23.4) | 176 (21.7) | |
| Not at all | 147 (18.1) | 111 (13.7) | 153 (18.9) | |
| Sex | | | | |
| Men | 373 (46.0) | - | - | |
| Women | 438 (54.0) | - | - | |
| Age (years) | | | | < 0.01 |
| 18–34 | 107 (13.2) | 87 (10.7) | 67 (8.3) | |
| 35–44 | 160 (19.7) | 148 (18.3) | 122 (15.0) | |
| 45–54 | 185 (22.8) | 187 (23.1) | 201 (24.8) | |
| 55–64 | 183 (22.6) | 182 (22.4) | 177 (21.8) | |
| 65–74 | 136 (16.8) | 151 (18.6) | 170 (21.0) | |
| $\geq$ 75 | 40 (4.9) | 56 (6.9) | 74 (9.1) | |
| Country of birth | | | | |
| France | 700 (86.3) | - | - | |
| Other countries | 111 (13.7) | - | - | |
| Education | | | | |
| < baccalaureate | 289 (35.6) | - | - | |
| baccalaureate to baccalaureate + 2 | 263 (32.4) | - | - | |
| $\geq$ baccalaureate + 3 | 259 (31.9) | - | - | |
| Number of people in the household | | | | 0.33 |
| 1 | 162 (20.0) | 161 (19.9) | 166 (20.5) | |
| 2 | 278 (34.3) | 296 (36.5) | 309 (38.1) | |
| 3 | 150 (18.5) | 150 (18.5) | 137 (16.9) | |
| $\geq$ 4 | 221 (27.3) | 204 (25.1) | 199 (24.5) | |
| Smoking habits | | ' | | 0.50 |
| Ex-smoker | 211 (26.1) | 200 (24.7) | 224 (27.6) | |
| Occasional/daily smoker | 175 (21.6) | 171 (21.1) | 157 (19.4) | |
| Non-smoker | 424 (52.3) | 440 (54.2) | 430 (53.0) | |
| Alcohol consumption | | | | 0.13 |
| Never | 207 (25.8) | 236 (29.1) | 245 (30.2) | |
| Light drinker | 425 (53.0) | 420 (51.8) | 409 (50.5) | |
| Moderate or heavy drinker | 170 (21.2) | 155 (19.1) | 156 (19.3) | |
| Sport activities | | | | 0.29 |
| Yes | 443 (54.6) | 444 (54.8) | 464 (57.2) | |
| No | 368 (45.4) | 367(45.2) | 347 (42.8) | |

(*Continued*)

**Table 1.** (Continued)

| Variables | T0 | T1 | T2 | p-value[1] |
|---|---|---|---|---|
| Income per CU[2] (per month) | | | | 0.04 |
| < 1550 euros | 294 (36.3) | 281 (34.7) | 237 (29.2) | |
| 1550 to 2750 euros | 368 (45.4) | 400 (49.3) | 440 (54.3) | |
| > 2750 euros | 149 (18.4) | 130 (16.0) | 134 (16.5) | |
| Noise sensitivity | | | | 0.48 |
| Highly noise sensitive | 239 (29.5) | 258 (31.8) | 243 (30.0) | |
| Not highly noise sensitive | 572 (70.5) | 553 (68.2) | 568 (70.0) | |
| Roof or windows insulation | | | | < 0.01 |
| Yes | 759 (93.8) | 808 (99.6) | 791 (97.5) | |
| No | 50 (6.2) | 3 (0.4) | 20 (2.5) | |
| Satisfaction with the living environment (score) | | | | < 0.01 |
| Mean (standard deviation) | 6.1 (2.6) | 6.4 (2.6) | 6.5 (2.5) | |

[1]p-value calculated using Cochran-Mantel-Haenszel chi-squared test

[2]Insee definition of CU: the first adult counts for 1 CU, the other people aged 14 or over for 0.5 CU, and the children under 14 years for 0.3 CU.

Fig 2 shows the results of SEM for the relationship between aircraft noise levels, aircraft noise annoyance and SRHS in longitudinal setting. At T0, the direct association between noise levels and poor /fair SRHS was significant (path coefficient: 0.08, p = 0.03). Aircraft noise levels seemed to affect SRHS through annoyance. Aircraft noise levels were associated with aircraft noise annoyance (path coefficient: 0.35, p <0.001), which in turn appeared associated with poor/fair SRHS (path coefficient: 0.06, p = 0.09). SRHS at T0 was associated with aircraft noise annoyance at T1 (path coefficient: 0.08, p <0.001).

Fig 3A shows the distribution of the final individual weights $\prod_{t=0}^{2} w_i^A(t) \times w_i^M(t)$ used to estimate the parameters of M1 model and Fig 3B shows the distribution of the final individual weights $\prod_{l=0}^{t} w_i^A(l), = 0, 1, 2$ used to estimate the parameters of M2 model. The mean of the final individual weights was close to 1 with relatively low standard deviations. For example, the mean of the final individual weights in the Fig 3A was 1.05 with a standard error equal to 1.18. Extreme weights remained relatively lower with respect to the stabilization (final individual weights in the Fig 3A ranged from $1.64 \times 10^{-8}$ to 16.50).

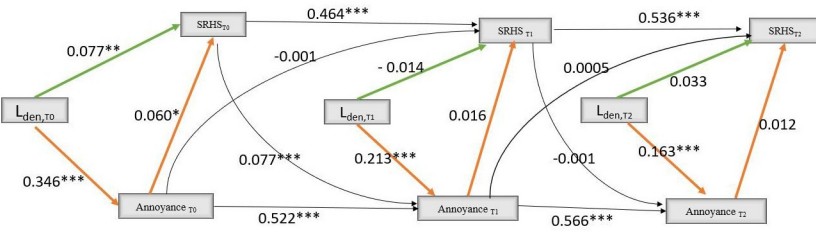

*p<=0.1, **p<=0.05    ***p<=0.001

**Fig 2. Results of structural equations models for the relationship between aircraft noise levels, aircraft noise annoyance and poor/fair self-reported health status.** Green arrows for direct effect, orange arrows for indirect effect and dotted arrows for confounding effect. Some arrows were omitted when no direct effect from a variable on another was hypothesized.

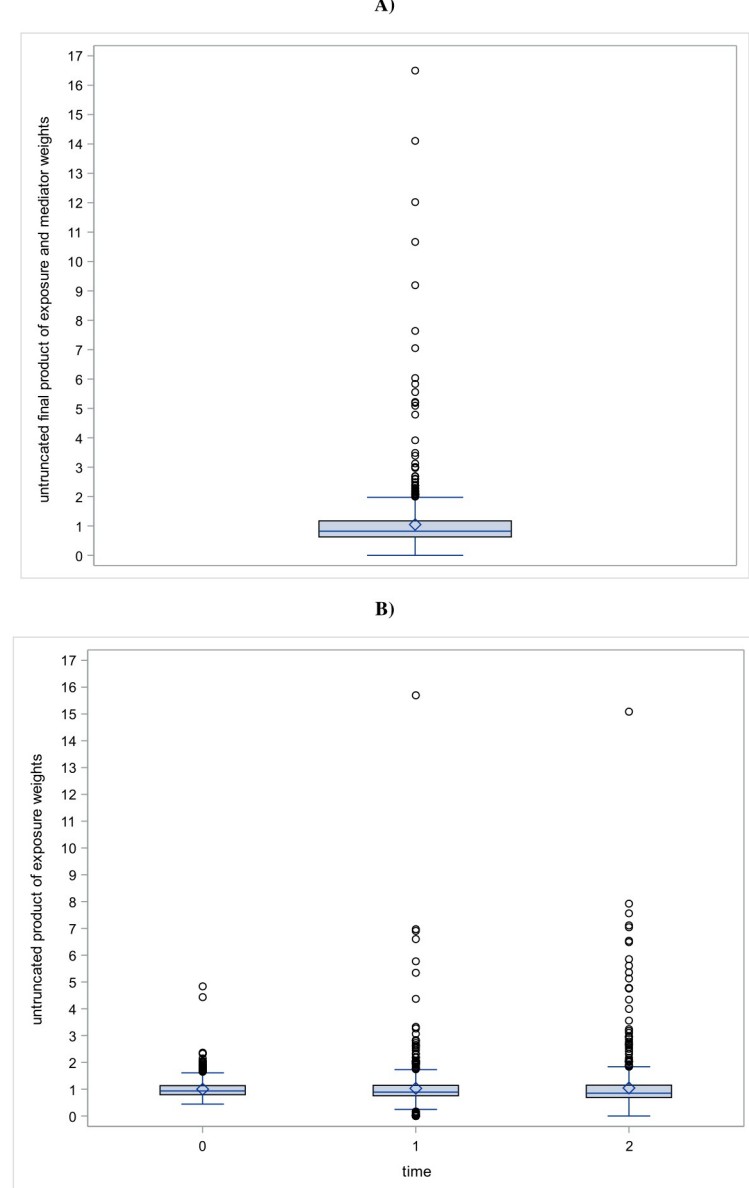

**Fig 3. Distributions of stabilized weights.** (A) product of exposure and mediators' stabilized weights, (B) product of exposure stabilized weights history for the three times.

Table 2 shows the estimates of the total interventional effect, the direct interventional effect, the indirect interventional effect of aircraft noise levels on SRHS and their confidence intervals. The ORs corresponding to the total interventional effect and comparing both intermediate aircraft noise categories (50–54 dBA and 55–59 dBA) to the reference category (<50 dBA), were not significant ($OR_{poor/fair\_SHRS}$ = 1.08 (95%CI: 0.90 to 1.45) and $OR_{poor/fair\_SHRS}$ = 1.16 (95%CI: 0.84 to 2.03), respectively). Only the OR corresponding to the total interventional effect and comparing the highest aircraft noise category (≥60 dBA) to the reference category (<50 dBA) was significant ($OR_{poor/fair_S HRS}$ = 1.25 (95%CI: 1.06 to 2.08)). The higher the noise category, the greater the total interventional effect seemed to be. For both intermediate noise

**Table 2. Causal interventional effects of aircraft noise levels in terms of odds ratio (OR) for poor/fair self-reported health status.**

| | 50–54 vs < 50 dBA | 55–59 vs < 50 dBA | ≥ 60 vs < 50 dBA |
|---|---|---|---|
| | OR (95%CI[1]) | OR (95% CI[1]) | OR (95%CI[1]) |
| Total interventional effect | 1.08 (0.90 to 1.45) | 1.16 (0.84 to 2.03) | 1.25 (1.06 to 2.08) |
| Direct interventional effect | 1.03 (0.82 to 1.37) | 1.05 (0.71 to 1.76) | 1.08 (0.79 to 1.80) |
| Indirect interventional effect | 1.05 (0.98 to 1.22) | 1.10 (0.96 to 1.46) | 1.16 (1.03 to 1.52) |
| Mediated proportion[2] | 63% | 64% | 66% |

[1]Confidence intervals obtained with a bootstrap of 7,500 resamples (convergence rate: 3,804/7,500) using the bootstrap bias correction method.

[2] Mediated proportion = = $\log(OR_{IIE})$/$\log(OR_{TIE}$ where IIE corresponds to indirect interventional effect and TIE to total interventional effect.

categories, the indirect interventional effect was not significant but greater than one ($OR_{poor/fair\_SHRS}$ = 1.05 (95%CI: 0.98 to 1.22) and $OR_{poor/fair\_SHRS}$ = 1.10 (95%CI: 0.96 to 1.46), respectively). For the highest aircraft noise category (≥60 dBA), the indirect interventional effect of aircraft noise levels on SRHS through aircraft noise annoyance was significant ($OR_{poor/fair\_SHRS}$ = 1.16 (95%CI: 1.03 to 1.52)): 66% of aircraft noise's effect on poor/fair SRHS was mediated by aircraft noise annoyance. The mediated proportion remained relatively similar in all noise categories.

The estimates from the sensitivity analysis using marginal models where the effect of aircraft noise levels on SRHS was not cumulative were of similar magnitude to those from the main analyses without reaching significance (S1 Table). The OR corresponding to the total interventional effect and comparing the highest aircraft noise category (≥ 60 dBA) to the reference category (<50 dBA) was $OR_{poor/fair\_SHRS}$ = 1.19 (95%CI: 0.78 to 2.61). About 65% of this total effect was mediated by aircraft noise annoyance ($OR_{poor/fair\_SHRS}$ = 1.12 (95%CI: 0.89 to 1.59)) (S1 Table).

The results from the sensitivity analysis considering aircraft noise annoyance (the mediator) as a binary variable (highly annoyed versus not highly annoyed) showed a slight increase in the effects. The OR corresponding to the total interventional effect of aircraft noise levels on SRHS and comparing the highest aircraft noise category (≥ 60 dBA) to the reference category (<50 dBA) was $OR_{poor/fair\_SHRS}$ = 1.33 (95%CI: 0.94 to 3.17), but it did not reach statistical significance. About 70% of this total effect was mediated by aircraft noise annoyance ($OR_{poor/fair\_SHRS}$ = 1.22 (95%CI:1.00 to 2.15)) (S1 Table).

The results from the sensitivity analysis using truncated weights were similar to those obtained in the main analyses (S1 Table).

## Discussion

This study investigated the mediating role of aircraft noise annoyance in the effect of aircraft noise levels on SRHS using a causal mediation analysis. The interventional effects (total, direct and indirect mediated by aircraft noise annoyance) of aircraft noise levels on SRHS were estimated with marginal structural models using data from the DEBATS longitudinal study conducted in France. In contrast to conventional approaches, marginal structural models allow causal effects to be estimated using inverse probability weighting mimicking a randomized trial in order to better control time-varying confounding factors. This study is the first to use this methodology to assess the effect of aircraft noise on SRHS and to investigate the mediating role of aircraft noise annoyance in this effect.

The results showed that the higher the noise category, the greater the total interventional effect seemed to be. Some previous studies using standard statistical methods also suggested an association between aircraft noise exposure and poor/fair SRHS [2, 3, 6] or poor health-related quality of life [4] for a 10 dBA increase in aircraft noise levels.

The present study showed no significant direct effect of aircraft noise levels on SRHS, but an indirect effect through annoyance. This indirect effect increased as aircraft noise levels increased, but the corresponding mediated proportion remained similar in the three highest noise levels compared to the reference category, around 65%. This result suggests importance of aircraft noise annoyance in the causal effect of exposure to aircraft noise levels on poor/fair SRHS. These results confirm those of previous longitudinal and cross-sectional studies which have attempted to identify the adverse causal effects of noise levels on health using standard structural equation models (SEM). Indeed, the NORAH longitudinal study conducted in Germany found a similar indirect effect of aircraft noise on mental well-being [8] and on depression diagnosis [20] through aircraft noise annoyance. Results from cross-sectional studies using SEM also suggested a mediating role of noise annoyance in the effect of noise on health, such as self-reported health [15], and general mental health [17, 18]. However, these studies failed to draw any conclusions about causality due to their cross-sectional design.

The main strength of this study relates to its longitudinal design and the use of causal inference methods. These approaches enable mediation analysis to be performed for time-varying exposure and mediator, even in the presence of potential mediator-outcome confounders affected by prior exposure to aircraft noise. To better control for bias, many potential confounding factors were included in the models. The results from the sensitivity analysis after additional adjustment for pre-existing diseases remain similar, with only a slight reduction in direct and total effects (S1 Table). An inverse causality of poor health condition on noise annoyance was observed in the NORAH longitudinal study: people with altered mental well-being [8] or with a diagnosis of depression [20] would be more annoyed by aircraft noise. Similar result was found in the present study: aircraft noise annoyance at inclusion (T0) was associated with SRHS at T0, which in turn was associated with aircraft noise annoyance at first follow-up (T1) (Fig 2). This potential inverse causality was considered by including previous SRHS values at T0 and T1 as confounders in the estimation of the mediator (aircraft noise annoyance) weight at T1 and T2 respectively. Stabilized individual weights were used, leading to a more efficient estimation of the parameters of marginal structural models [33]. To assess potential biases associated with extreme individual weights, weights truncated at the 5th and 95th percentiles were used in sensitivity analysis. The results did not differ from those obtained in the main analysis (S1 Table).

One limitation of this study is the potential misclassification in aircraft noise levels and aircraft noise annoyance. In the DEBATS study, aircraft noise levels were estimated outside the participants' home. No information was available on their exposure indoors or during the day when they were away from home. Furthermore, the use of aircraft noise maps from 2003 for Lyon-Saint-Exupéry airport and 2004 for Toulouse-Blagnac airport (as more recent noise maps were not available) may underestimate actual aircraft noise levels between 2013 and 2017 for these two airports. But this bias seems very unlikely. Indeed, the number of aircraft movements did not change over time for the three airports included in the DEBATS study (Fig A in S1 Fig). In addition, using the noise maps available for Paris-Charles de Gaulle airport between 2012 and 2017, the exposures to aircraft noise levels of participants who had not moved during this period did not differ from those observed in 2013 (Fig B in S1 Fig). In another study, Bruitparif, the noise observatory for the Ile-de-France region, showed that for Paris-Charles de Gaulle airport, changes in traffic alone lead to relatively small variations of around +/- 0.4 dB [36]. Potential changes in aircraft noise levels would be likely lower than 5

dBA, which corresponds to the range of noise classes within which participants are grouped in this study. A classification bias when assessing the aircraft noise level cannot be ruled out, but this potential bias is likely non-differential as it is not related to the SRHS outcome and would lead to an underestimation of the real effects of aircraft noise levels in this study. Aircraft noise annoyance was assessed using a single standardized question validated by ICBEN [29], with a 5-point verbal response scale, which may not be sufficient to capture noise annoyance in its various aspects. A multi-item noise annoyance scale could provide a comprehensive complement for noise annoyance assessment [37]. SRHS was assessed using a unique subjective question. Response to this question by the participant could be filtered through their perceptions and could thus not reflect their real multidimensional health. In addition, as DEBATS is a study on the health effects of aircraft noise, participants' responses could be prone to response bias, particularly with regards to self-reporting of SRHS and annoyance.

Potential model misspecification bias cannot be ruled out. The main analysis assumed a cumulative effect of aircraft noise levels and aircraft noise annoyance on SRHS at the end of the follow-up. However, the specifications may be more complex, for example interaction between the cumulative effect of noise levels and the cumulative effect of annoyance on SRHS in the M1 model may occur. Indeed, in the NORAH longitudinal study, increasing aircraft noise levels appeared to interact with annoyance on altered mental well-being [8]. In the presence of an interaction, the identification of interventional effects becomes more complex [25]. Therefore, a model misspecification bias cannot be excluded in this study. Nevertheless, alternative specification of the marginal structural models M1 and M2 were examined. Assuming, for example, non-cumulative effects, the estimated interventional effects remained similar, but the confidence intervals were less precise due to the larger number of estimated parameters (S1 Table). In addition, the IPW method we used only controls for measured confounders and not for unmeasured confounders, resulting in potential bias. Thus, the models used in this study are valid under the strong assumption that $L(t)$ were sufficient to control for confounding bias, and that they were well characterized by the information collected in the questionnaire. Unfortunately, as in all observational studies, this strong assumption cannot be verified from the data. Further high computational work could be done for assessing the impact of a violation of this assumption on the results [34].

At the same time point, an effect of annoyance on SRHS was assumed (Fig 1) since participants were asked to report their noise annoyance in the 12 months preceding the interview and their SRHS on the day of the interview. A reciprocal effect of SRHS on the reporting of annoyance, inherent to the subjective nature of the exposure and of the outcome, could not be ruled out and could therefore induce misspecification bias. This potential reciprocal effect was partially addressed through a time-varying design at later time points (Fig 1).

Ideally, from a statistical standpoint, considering noise levels as continuous variables seems more favorable to avoid losing information typically due to categorization. Nevertheless, the IPTW estimation method used in this study requires a good understanding of the distribution of aircraft noise levels [38]. This distribution is often assumed to be a normal distribution. However, in this study the distribution diverged from a normal distribution. In the case of unknown distribution, categorization of the variable remains the alternative solution [38]. We have therefore opted for a categorization into four classes corresponding to the different aircraft noise contours of the three airports included in the DEBATS study.

Almost 65% of the effect of aircraft noise on fair/poor SRHS is mediated by noise annoyance. The mediation analysis presented in this study was limited to considering aircraft noise annoyance as the only mediator. Factors associated with annoyance, such as sleep disturbance and stress, could mediate the causal effect of aircraft noise levels on health, and explain part of the remaining 35% of the total effect. Indeed, deterioration in sleep quality and sleep

interruptions, as well as noise annoyance, are considered to be among the possible key variables in the causal pathway of noise-induced cardiovascular and metabolic diseases [39]. The causal relationship between aircraft noise annoyance and sleep disturbance remains unclear, and a reverse causal effect is also plausible [40]. It is also known that prolonged noise-induced physiological activation can trigger biological risk factors, such as altered blood pressure, which are directly linked to long-term health effects [41]. Therefore, a mediation analysis jointly assessing several mediators may provide a better identification of the causal effect of noise levels on health. However, the causal approach used in this study remains complex to implement for a mediation analysis including multiple mediators. Finally, other environmental risk factors, such as air pollution, may also contribute to subjective health impairment. The potential interaction between air pollution and aircraft noise, which share the same sources, on subjective health was not explored in this study and will be the subject of future work.

## Conclusion

This study supports the deleterious effect of aircraft noise on SRHS. The results highlight the important mediating role of aircraft noise annoyance in the causal pathway from exposure to aircraft noise to poor/fair SRHS. Using causal inference methods provided better control of confounding factors and a more accurate estimation of causal effects. Considering other potential mediators, including sleep disturbances, in the joint mediation analysis may help to better understand the effects of aircraft noise exposure on health.

## Supporting information

**S1 Table. Causal interventional effects of aircraft noise levels on poor/fair self-reported health, sensitivity analyses.**
(DOCX)

**S1 Fig. Trends in the number of aircraft movements and distribution of aircraft noise levels.**
(DOCX)

## Acknowledgments

The Airport Pollution Control Authority (Acnusa) requested Gustave Eiffel University to carry out this study. The authors would like to thank them for their confidence. The authors are grateful to all the participants in the study and their interviewers.

The authors also thank Paris Airports and the French Civil Aviation Authority for providing noise exposure maps, and are also grateful to Inès Khati and Marie Lefèvre for their participation in the implementation of the study.

The authors would also like to warmly thank Lauren Henry for her careful proofreading of the manuscript.

This work has been conducted in the framework of CeLyA (Lyon Center for Acoustics, ANR-10-LABX-0060).

## Author Contributions

**Conceptualization:** Anne-Sophie EVRARD.

**Data curation:** Minon'tsikpo kossi KODJI, Lise GIORGIS-ALLEMAND.

**Formal analysis:** Minon'tsikpo kossi KODJI, Émilie LANOY, Anne-Sophie EVRARD.

**Funding acquisition:** Anne-Sophie EVRARD.

**Methodology:** Minon'tsikpo kossi KODJI.

**Supervision:** Lise GIORGIS-ALLEMAND, Sylviane LAFONT, Émilie LANOY, Anne-Sophie EVRARD.

**Writing – original draft:** Minon'tsikpo kossi KODJI, Anne-Sophie EVRARD.

**Writing – review & editing:** Minon'tsikpo kossi KODJI, Lise GIORGIS-ALLEMAND, Sylviane LAFONT, Émilie LANOY, Anne-Sophie EVRARD.

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
