## [Decision Letter · Decision Letter 0]

18 Apr 2024

PONE-D-24-06019Effects of aircraft noise exposure on self-reported health through aircraft noise annoyance: causal mediation analysis in the DEBATS longitudinal study in FrancePLOS ONE

Dear Dr. KODJI,

Thank you for submitting your manuscript to PLOS ONE. After careful consideration, we feel that it has merit but does not fully meet PLOS ONE’s publication criteria as it currently stands. Therefore, we invite you to submit a revised version of the manuscript that addresses the points raised during the review process.

We look forward to receiving your revised manuscript.

Kind regards,

Yuan Zhang, PhD

Academic Editor

PLOS ONE

   "The present study was supported by funds from the French Ministry of Health, the French Ministry of the Environment, the French Civil Aviation Authority, the Airport Pollution Control Authority (Acnusa) and the Gustave Eiffel University. The authors would like to thank them for their kind assistance."

4. For studies involving third-party data, we encourage authors to share any data specific to their analyses that they can legally distribute. PLOS recognizes, however, that authors may be using third-party data they do not have the rights to share. When third-party data cannot be publicly shared, authors must provide all information necessary for interested researchers to apply to gain access to the data. (https://journals.plos.org/plosone/s/data-availability#loc-acceptable-data-access-restrictions) 

Reviewers' comments:

Reviewer's Responses to Questions

**Comments to the Author**

1. Is the manuscript technically sound, and do the data support the conclusions?

Reviewer #1: Partly

Reviewer #2: Yes

Reviewer #3: Partly

2. Has the statistical analysis been performed appropriately and rigorously? 

Reviewer #1: Yes

Reviewer #2: Yes

Reviewer #3: Yes

3. Have the authors made all data underlying the findings in their manuscript fully available?

Reviewer #1: Yes

Reviewer #2: Yes

Reviewer #3: Yes

4. Is the manuscript presented in an intelligible fashion and written in standard English?

Reviewer #1: Yes

Reviewer #2: Yes

Reviewer #3: Yes

5. Review Comments to the Author

Reviewer #1: Summary: Based on longitudinal data of three waves with interviews between 2013 and 2017 of 881 participants, causal pathways from aircraft noise exposure to self-reported health status are studied using marginal structural models with inverse probability weighting. It was found that the indirect effect through noise annoyance was dominant over the direct pathway through noise exposure.

Main impression: The DEBATS study from France has already produced several interesting publications with important insights in the effect chains from aircraft noise exposure to detrimental health effects. The current analysis shows a methodically interesting design and contributes to the discussion about the importance of indirect pathways.

Personally, I see one fundamental weakness of the study, namely the fact that the exposure data of two of the three airports was not corresponding to the survey years and in addition that it was assumed to be constant over all three years. With aircraft noise exposure being one of the two core input parameters for the modelling, the data quality seems not good enough for a longitudinal study and to draw these conclusions. The results of the structural equations models as given in Figure 2 support my concern as the association between Lden and SRHS changes sign from year to year, which seems rather unlikely to me.

Therefore, I suggest to refine the exposure modelling or at least to justify the assumption that the aircraft noise exposure of the years 2013, 2015 and 2017 was identical to 2003/2004. In order to do that, the following steps should be taken: (a) compare the number of movements, ideally separately for day, evening and night, (b) insure that the fleet mix remained stable (no introduction of new aircraft types) and (c) that the distribution of routes remained stable. This comparison should also be done for Paris-Charles de Gaulle airport and the outcome should be compared with the available aircraft noise contours of Paris-Charles de Gaulle.

In parallel, it would be interesting to rerun the models using exclusively the data of Paris, which exhibit better quality input data.

Comments in detail:

L. 48 This is the first time SRHS is used in the text. Therefore I suggest to use the full expression including "status".

L. 77 "Tchetgen" comes twice

L. 100ff As discussed above…

L. 132 Please indicate how the quality of sound insulation was determined.

L. 154 The sentence ends with a bracket instead of a point. The bracket is closed on line 161.

L. 157 This should be part of the caption, but looks like normal text.

L. 167 The caption of Figure 1 is repeated and should be deleted.

L. 210 "et" instead of "and"?

L. 251 "Aircraft noise levels seems affected SRHS through annoyance." Please revise this sentence.

L. 266 Fig. 3, at least in the way it is given at the end of the document, shows a bad resolution and is hard to read.

L. 371 "But this bias seems very unlikely. … showed no significant difference in exposure levels." As previously mentioned, this conclusion should be justified further. How do you assess whether a change in exposure is significant or not?

Reviewer #2: This is a very interesting study aimed at determining the causal mediating role of aircraft noise annoyance in the association between aircraft noise levels and self-reported health. The authors are experienced in the field, as they have previously published interesting data from the DEBATS study. This is a valuable contribution to the research field with very novel and robust methodology, as there is an ongoing discussion about the role of noise annoyance. I guess the major weakness of this study is the outcome, as the health status is evaluated by only one item. I have only minor comments:

• Abstract: Could you provide some odds ratios (ORs) so that one can get a sense of the magnitude of effects?

• Aircraft noise exposure: Does it make sense to present the results with continuous aircraft noise levels as well?

• With the term “gender” I assume you mean “sex”.

• Do you have data on pre-existing diseases or newly developed diseases at follow-up and medication intake?

• Is there any chance to “validate” the outcome with more objective or subjective data?

• Could you propose what might account for the remaining 35% when 65% of the effect is mediated by noise annoyance? From my understanding, there was no direct effect of aircraft noise on health status. Does this imply that only when annoyance occurs as a response, it influences health status? In other words, would it be correct to state that without annoyance, there is no adverse impact on health?"

Reviewer #3: This study uses a longitudinal cohort study in France along with estimates of participants’ aircraft noise exposures to investigate mediation of effects on self-reported health through self-reported annoyance. The design is innovative for its application of a causal mediation framework to help elucidate – how transportation noise, particularly aircraft noise, may impact health, and through which mechanisms. While the design is robust and uniquely applied, there are some important limitations that the authors do not address thoroughly.

The authors mention that there are unmeasured mediators that are likely to play a role in health and hint at the likelihood for unmeasured confounders. And yet they assume they have correctly specified the model. These two ideas are incongruous. I would suspect that, for the reasons the authors mentioned, the model is indeed misspecified (though no major fault of the authors – the right data is rarely, if ever, actually available, and such efforts are inherently idealistic but laudable, no less). And that is why greater nuance in the reporting of the results needs to be added in key places throughout the manuscript (abstract, discussion, conclusion). Adding in these factors may change the indirect effects of annoyance dramatically. Lines 337-340 are an example of overly strong language that needs to be toned back. This exercise still has great utility and demonstrates a recipe for other researchers to follow, but there are important assumptions in causal inference as there are with other techniques that are not given enough attention, in my opinion.

One subjective question on SRHS is a coarse and crude metric upon which to base a study. Granted, the design is thoughtful; but, it cannot be overstated that this is a single dimension of health perceived by the participant measured in a highly subjective fashion. Presumably, participants are filtering responses through their perceptions, and many of these are fundamentally mental processes. The lack of objectivity does not allow the researchers to understand the dimensions of health, whether even physical or mental, much less the other subdimensions of health. This major limitation needs to be added to the limitations section, at minimum.

Relatedly, I think there is decent potential for reverse causality despite the innovative design that does include the relationship between SRHS and subsequent annoyance. However, I would find it somewhat difficult to believe that SRHS and annoyance are not correlated and intertwined at the same time point, particularly at baseline. The orange arrow in Fig 1 from annoyance to SRHS at baseline is a strong assumption. How can the authors rule out reverse causality here? I could see how perceived SRHS could make increase annoyance. I think this point is major and not minor. The authors partially address this through their time-varying design at later time points; but I am failing to see how the self-reported nature of similar concepts (health and annoyance) are not fundamentally related. If they share a common ancestor from the causal inference perspective, there would be bias introduced by improper/lack of conditioning. I think this point is nuanced but important, and not as well articulated as it should be.

The DEBATS study is a study on the health effects of aircraft noise. As such, participants are prone to response bias, particularly in the self-reporting of SRHS and annoyance. That should be mentioned as an additional limitation and considered throughout the study.

It’s too bad that 2/3 airports had outdated and non-time varying aircraft noise exposure estimates. Did I miss somewhere the changes in aircraft noise exposures among participants over time at CDG? If not, I think it may be useful to show, at minimum, the changes in exposures for participants living around CDG that did not move over the study period by year, perhaps in the supplement. It would enable crude justification that the patterns at the other 2 airports were similar. The authors could consider a quantitative bias analysis if they thought prudent.

Why did the author not consider adding analyses of dichotomized aircraft noise exposures at different cut points, and only investigate it as categorical? Is an analysis of continuous exposure possible in this framework?

The same footnote for Fig 1 needs to be included for Fig 2.

Lines 251-252: typo/incomplete sentence.

Table 2: there should be no acronyms in the title (e.g., SRHS) to help it be “standalone”.

6. PLOS authors have the option to publish the peer review history of their article (what does this mean?). If published, this will include your full peer review and any attached files.

Reviewer #1: No

Reviewer #2: No

Reviewer #3: No

---

## [Author Response · Author response to Decision Letter 0]

16 May 2024

Response to Reviewers

The co-authors would like to thank the reviewers for their comments, which give them the opportunity to clarify certain aspects. 

Files named Response to Reviewers contents all response to reviewer's comments. The lines indicated in the response to the reviewers correspond to the lines in the revised versions of the manuscript with track changes.

---

## [Decision Letter · Decision Letter 1]

19 Jun 2024

PONE-D-24-06019R1Effects of aircraft noise exposure on self-reported health through aircraft noise annoyance: causal mediation analysis in the DEBATS longitudinal study in FrancePLOS ONE

Dear Dr. KODJI,

Thank you for submitting your manuscript to PLOS ONE. After careful consideration, we feel that it has merit but does not fully meet PLOS ONE’s publication criteria as it currently stands. Therefore, we invite you to submit a revised version of the manuscript that addresses the points raised during the review process.

We look forward to receiving your revised manuscript.

Kind regards,

Yuan Zhang, PhD

Academic Editor

PLOS ONE

Journal Requirements:

Reviewers' comments:

Reviewer's Responses to Questions

**Comments to the Author**

1. If the authors have adequately addressed your comments raised in a previous round of review and you feel that this manuscript is now acceptable for publication, you may indicate that here to bypass the “Comments to the Author” section, enter your conflict of interest statement in the “Confidential to Editor” section, and submit your "Accept" recommendation.

Reviewer #1: All comments have been addressed

Reviewer #2: All comments have been addressed

Reviewer #3: All comments have been addressed

2. Is the manuscript technically sound, and do the data support the conclusions?

Reviewer #1: Yes

Reviewer #2: Yes

Reviewer #3: Yes

3. Has the statistical analysis been performed appropriately and rigorously? 

Reviewer #1: Yes

Reviewer #2: Yes

Reviewer #3: Yes

4. Have the authors made all data underlying the findings in their manuscript fully available?

Reviewer #1: Yes

Reviewer #2: Yes

Reviewer #3: Yes

5. Is the manuscript presented in an intelligible fashion and written in standard English?

Reviewer #1: Yes

Reviewer #2: Yes

Reviewer #3: No

6. Review Comments to the Author

Reviewer #1: I thank the authors for thoroughly taking up and discussing my concerns. In my perception, the paper is now ready for being published.

Reviewer #2: Q3- With the term “gender” I assume you mean “sex”.

Not really. The gender was recorded by the interviewer without asking the participant. It may therefore differ from the biological sex.

How did the interviewer exactly recorded gender?

Q4- Do you have data on pre-existing diseases or newly developed diseases at follow-up and medication intake?

We have information on the existence of certain pre-existing diseases such as cardiovascular disease, diabetes, cancer, diagnosed hypertension, or the use of antihypertensive medication or of other medication. Some of these have already been the subject of published articles and others will be the subject of future publications.

I think it might be useful to control for them - do you agree?

Reviewer #3: The authors have addressed my comments well. However, there are still many minor grammatical errors and I recommend a close re-reading to comb through the manuscript before the submit their final version.

7. PLOS authors have the option to publish the peer review history of their article (what does this mean?). If published, this will include your full peer review and any attached files.

Reviewer #1: **Yes: **Jean Marc Wunderli

Reviewer #2: No

Reviewer #3: No

---

## [Author Response · Author response to Decision Letter 1]

1 Jul 2024

The response to reviewers is uploaded as a separate file labeled 'Response to Reviewers'.

---

## [Editor Report · Decision Letter 2]

11 Jul 2024

Effects of aircraft noise exposure on self-reported health through aircraft noise annoyance: causal mediation analysis in the DEBATS longitudinal study in France

PONE-D-24-06019R2

Dear Dr. KODJI,

We’re pleased to inform you that your manuscript has been judged scientifically suitable for publication and will be formally accepted for publication once it meets all outstanding technical requirements.

Kind regards,

Yuan Zhang, PhD

Academic Editor

PLOS ONE
---

## [Editor Report · Acceptance letter]

15 Jul 2024

PONE-D-24-06019R2 

PLOS ONE

Dear Dr. KODJI, 

I'm pleased to inform you that your manuscript has been deemed suitable for publication in PLOS ONE. Congratulations! Your manuscript is now being handed over to our production team.

Kind regards, 

on behalf of

Professor Yuan Zhang 

Academic Editor

PLOS ONE